# Glyph: Fast and Accurately Training Deep Neural Networks on Encrypted Data

Qian Lou
louqian@iu.edu

Bo Feng
fengbo@iu.edu

Geoffrey C. Fox
gcf@indiana.edu

Lei Jiang
jiang60@iu.edu

Indiana University Bloomington

## Abstract

Because of the lack of expertise, to gain benefits from their data, average users have to upload their private data to cloud servers they may not trust. Due to legal or privacy constraints, most users are willing to contribute only their encrypted data, and lack interests or resources to join deep neural network (DNN) training in cloud. To train a DNN on encrypted data in a completely non-interactive way, a recent work proposes a fully homomorphic encryption (FHE)-based technique implementing all activations by *Brakerski-Gentry-Vaikuntanathan* (BGV)-based lookup tables. However, such inefficient lookup-table-based activations significantly prolong private training latency of DNNs.

In this paper, we propose, Glyph, a FHE-based technique to fast and accurately train DNNs on encrypted data by switching between TFHE (Fast Fully Homomorphic Encryption over the Torus) and BGV cryptosystems. Glyph uses logic-operation-friendly TFHE to implement nonlinear activations, while adopts vectorial-arithmetic-friendly BGV to perform multiply-accumulations (MACs). Glyph further applies transfer learning on DNN training to improve test accuracy and reduce the number of MACs between ciphertext and ciphertext in convolutional layers. Our experimental results show Glyph obtains state-of-the-art accuracy, and reduces training latency by $69\% \sim 99\%$ over prior FHE-based privacy-preserving techniques on encrypted datasets.

## 1 Introduction

Deep learning is one of the most dominant approaches to solving a wide variety of problems such as computer vision and natural language processing [1], because of its state-of-the-art accuracy. By only sufficient data, DNN weights can be trained to achieve high enough accuracy. Average users typically lack knowledge and expertise to build their own DNN models to harvest benefits from their own data, so they have to depend on big data companies such as Google, Amazon and Microsoft. However, due to legal or privacy constraints, there are many scenarios where the data required by DNN training is extremely sensitive. It is risky to provide personal information, e.g., financial or healthcare records, to untrusted companies to train DNNs. Federal privacy regulations also restrict the availability and sharing of sensitive data.

Recent works [2, 3, 4] propose cryptographic schemes to enable privacy-preserving training of DNNs. Private federated learning [4] (FL) is created to decentralize DNN training and enable users to train with their own data locally. QUOTIENT [3] takes advantage of multi-party computation (MPC) to interactively train DNNs on both servers and clients. Both FL and MPC require users to stay online and heavily involve in DNN training. However, in some cases, average users may not have strong interest, powerful hardware, or fast network connections for interactive DNN training [5]. To enable DNN training on encrypted data in a completely non-interactive way, a recent study presents the first fully homomorphic encryption (FHE)-based stochastic gradient descent technique [2], FHESGD.

During FHESGD, a user encrypts and uploads private data to an untrusted server that performs both forward and backward propagations on the encrypted data without decryption. After uploading encrypted data, users can simply go offline. Privacy is preserved during DNN training, since input and output data, activations, losses and gradients are all encrypted.

However, FHESGD [2] is seriously limited by its long training latency, because of its BGV-lookup-table-based *sigmoid* activations. Specifically, FHESGD builds a Multi-Layer Perceptron (MLP) with 3 layers to achieve $< 98\%$ test accuracy on an encrypted MNIST after 50 epochs. A mini-batch including 60 samples takes $\sim 2$ hours on a 16-core CPU. FHESGD uses the BGV cryptosystem [6] to implement stochastic gradient descent, because BGV is good at performing large vectorial arithmetic operations frequently used in a MLP. However, FHESGD replaces all activations of a MLP by *sigmoid* functions, and uses BGV table lookups [7] to implement a sigmoid function. A BGV table lookup in the setting of FHESGD is so slow that BGV-lookup-table-based *sigmoid* activations consume $\sim 98\%$ of the training time.

In this paper, we propose a FHE-based technique, Glyph, to enable fast and accurate training over encrypted data. Glyph adopts the logic-operation-friendly TFHE cryptosystem [8] to implement activations such as *ReLU* and *softmax* in DNN training. TFHE-based activations have shorter latency. We present a cryptosystem switching technique to enable Glyph to perform activations by TFHE and switch to the vectorial-arithmetic-friendly BGV when processing fully-connected and convolutional layers. By switching between TFHE and BGV, Glyph substantially improves the speed of privacy-preserving DNN training on encrypted data. At last, we apply transfer learning on Glyph to not only accelerate private DNN training but also improve its test accuracy. Glyph achieves state-of-the-art accuracy, and reduces training latency by $69\% \sim 99\%$ over prior FHE-based privacy-preserving techniques on encrypted datasets.

## 2 Background

**Threat Model**. Although an encryption scheme protects data sent to external servers, untrusted servers [1] can make data leakage happen. Homomorphic Encryption is one of the most promising techniques to enable a server to perform private DNN training [2] on encrypted data. A user sends encrypted data to a server performing private DNN training on encrypted data. After uploading encrypted data to the server, the user may go offline immediately.

**Fully Homomorphic Encryption**. A homomorphic encryption [9] (HE) cryptosystem encrypts plaintext $p$ to ciphertext $c$ by a function $\epsilon$. $c = \epsilon(p, k_{pub})$, where $k_{pub}$ is the public key. Another function $\sigma$ decrypts ciphertext $c$ back to plaintext $p$. $p = \sigma(c, k_{pri})$, where $k_{pri}$ is the private key. An operation $\star$ is *homomorphic*, if there is another operation $\circ$ such that $\sigma(\epsilon(x, k_{pub}) \circ \epsilon(y, k_{pub}), k_{priv}) = \sigma(\epsilon(x \star y, k_{pub}), k_{priv})$, where $x$ and $y$ are two plaintext operands. Each HE operation introduces a noise into the ciphertext. *Leveled* HE (LHE) allows to compute HE functions of only a maximal degree by designing a set of parameters. Beyond its maximal degree, LHE cannot correctly decrypt the ciphertext, since the accumulated noise is too large. On the contrary, *fully* HE (FHE) can enable an unlimited number of HE operations on the ciphertext, since it uses *bootstrapping* [6, 8] to "refresh" the ciphertext and reduce its noise. However, bootstrapping is computationally expensive. Because privacy-preserving DNN training requires an impractically large maximal degree, it is impossible to train a DNN by LHE. A recent work [2] demonstrates the feasibility of using FHE BGV to train a DNN on encrypted data.

**BGV, BFV, and TFHE**. Based on Ring-LWE (Learning With Errors), multiple FHE cryptosystems [8, 6], e.g., TFHE [8], BFV [10], BGV [6], HEAAN [11], are developed. Each FHE cryptosystem can more efficiently process a specific type of homomorphic operations. For instance, TFHE [8] runs combinatorial operations on individual slots faster. BFV [10] is good at performing large vectorial arithmetic operations. Similar to BFV, BGV [8] manipulates elements in large cyclotomic rings, modulo integers with many hundreds of bits. However, BGV has less scaling operations, and thus processes vectorial multiplications of ciphertexts faster [12, 13]. At last, HEAAN [11] supports floating point computations better. A recent work [14] demonstrates the feasibility of combining and switching between TFHE, BFV and HEAAN via homomorphic operations.

**Forward and Backward Propagation**. DNN training includes both forward and backward propagations. During forward propagation, the input data go through layers consecutively in the forward direction. Forward propagation can be described as $u_l = W_l d_{l-1} + b_{l-1}$ and $d_l = f(u_l)$, where $u_l$ is the neuron tensor of layer $l$; $d_{l-1}$ is the output of layer $l-1$ and the input of layer $l$; $W_l$ is the weight

| Operation | BFV (s) | BGV (s) | TFHE (s) |
|---|---|---|---|
| MultCC | 0.043 | 0.012 | 2.121 |
| MultCP | 0.006 | 0.001 | 0.092 |
| AddCC | 0.0001 | 0.002 | 0.312 |
| TLU | / | 307.9 | 3.328 |

Table 1: The latency comparison of FHE operations. **MultCC**: ciphertext $\times$ ciphertext. **MultCP**: ciphertext $\times$ plaintext. **AddCC**: ciphertext + ciphertext. **TLU**: table lookup.

| FC | Act | FC (s) | Act (s) |
|---|---|---|---|
| BFV | / | 9191 | / |
| BGV | BGV | 2891 | 114980 |
| TFHE | TFHE | 716800 | 65 |
| BFV | TFHE | 9209 | 84 |
| BGV | TFHE | 2909 | 82 |

Table 2: The comparison of mini-batch latency of various FHE-based private training. **FC**: fully-connected layer. **Act**: Activation.

tensor of layer $l$; $b_{l-1}$ is the bias tensor of layer $l-1$; and $f()$ is the forward activation function. We use $y$ and $t$ to indicate the output of a neural network and the standard label, respectively. An $L^2$ norm loss function is defined as $E(W, b) = \frac{1}{2}||y - t||_2^2$. Backward propagation can be described by $\delta_{l-1} = (W_l)^T \delta_l \circ f'(u_l)$, $\nabla W_l = d_{l-1}(\delta_l)^T$, and $\nabla b_l = \delta_l$, where $\delta_l$ is the error of layer $l$ and defined as $\frac{\partial E}{\partial b_l}$; f'() is the backward activation function; $\nabla W_l$ and $\nabla b_l$ are weight and bias gradients.

**BGV-based FHESGD**. BGV-based FHESGD [2] trains a 3-layer MLP using *sigmoid* activations, and implements *sigmoid* by a lookup table. However, lookup-table-based *sigmoid* activations significantly increase mini-batch training latency of FHESGD. As Figure 1 shows, with an increasing bitwidth of each entry of a BGV-based *sigmoid* lookup table, test accuracy of FHESGD improves and approaches 98%, but its activation processing time, i.e., *sigmoid* table lookup latency, also significantly increases and occupies > 98% of mini-batch training latency.

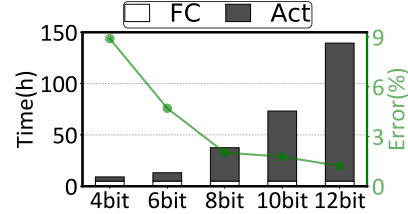
Figure 1: FHESGD-based MLP.

**THFE-based Training**. It is possible to fast and accurately implement homomorphic activations including *ReLU* and *softmax* of private training by TFHE, since the TFHE cryptosystem processes combinatorial operations on individual slots more efficiently. Table 1 compares latencies of various homomorphic operations implemented by BGV, BFV and TFHE. Compared to BGV, TFHE shortens table lookup latency by $\sim 100\times$, and thus can implement faster activation functions. However, after we implemented private DNN training by TFHE, as Table 2 exhibits, we found although (TFHE) homomorphic activations take much less time, the mini-batch training latency substantially increases, because of slow TFHE homomorphic MAC operations. As Table 1 shows, compared to TFHE, BGV [8] demonstrates $17\times \sim 30\times$ shorter latencies for a variety of vectorial arithmetic operations such as a multiplication between a ciphertext and a ciphertext (MultCC), a multiplication between a ciphertext and a plaintext (MultCP), and an addition between a ciphertext and a ciphertext (AddCC). Therefore, if we implement activation operations by TFHE, and compute vectorial MAC operations by BGV, private DNN training obtains both high test accuracy and short training latency.

**BFV-TFHE Switching**. Although a recent work [14] proposes a cryptosystem switching technique, Chimera, to homomorphically switch between TFHE and BFV, we argue that compared to BFV, BGV can implement faster private DNN training. As Table 1 shows, BGV computes MultCPs and MultCCs faster than BFV, because it has less scaling operations [12, 13]. In this paper, we propose a new cryptosystem technique to enable the homomorphic switching between BGV and TFHE. Though BFV supports faster AddCCs, we show our Glyph achieves much shorter training latency than Chimera in Section 5.1, since private MultCPs and MultCCs dominate training latency of DNNs.

## 3  Glyph

### 3.1  TFHE-based Activations

To accurately train a FHE-based DNN, we propose TFHE-based homomorphic *ReLU* and *softmax* activation units. We construct a *ReLU* unit by TFHE homomorphic gates with bootstrapping, and build a *softmax* unit by TFHE homomorphic multiplexers.

**Forward ReLU**. The forward *ReLU* of the $i_{th}$ neuron in layer $l$ can be summarized as: if $u_l^i \geq 0$, $d_l^i = ReLU(u_l^i) = u_l^i$; otherwise, $d_l^i = ReLU(u_l^i) = 0$, where $u_l^i$ is the $i_{th}$ neuron in layer $l$. A 3-bit TFHE-based forward *ReLU* unit is shown in Figure 2(a), where we first set the most significant bit (MSB) of $d_l^i$, i.e., $d_l^i[2]$, to 0, so that $d_l^i$ can be always non-negative. We then get the negation of the MSB of $u_l^i$, $\overline{u_l^i[2]}$, by a TFHE homomorphic NOT gate that even does not require bootstrapping [9].

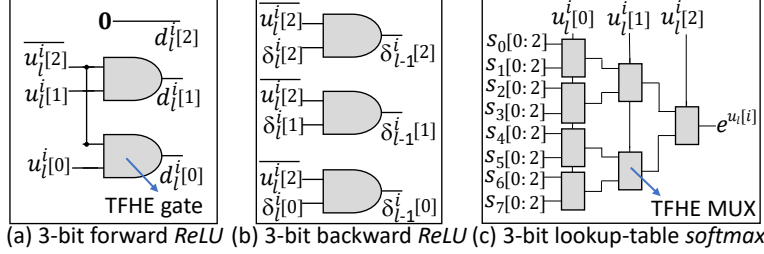

(a) 3-bit forward *ReLU*    (b) 3-bit backward *ReLU*    (c) 3-bit lookup-table *softmax*

Figure 2: TFHE-based activations.

If $u_l^i$ is positive, $\overline{u_l^i[2]} = 1$; otherwise $\overline{u_l^i[2]} = 0$. At last, we compute $d_l^i[0:1]$ by ANDing each bit of $u_l^i$ with $\overline{u_l^i[2]}$. So if $u_l^i$ is positive, $d_l^i = u_l^i$; otherwise $d_l^i = 0$. An $n$-bit forward *ReLU* unit requires 1 TFHE NOT gate without bootstrapping and $n-1$ TFHE AND gates with bootstrapping.

**Backward iReLU**. The backward *iReLU* for the $i_{th}$ neuron in layer $l$ can be described as: if $u_l^i \geq 0$, $iReLU(u_l^i, \delta_l^i) = \delta_{l-1}^i = \delta_l^i$; otherwise, $iReLU(u_l^i, \delta_l^i) = \delta_{l-1}^i = 0$, where $\delta_l^i$ is the $i_{th}$ error of layer $l$. The backward *iReLU* takes the $i_{th}$ error of layer $l$, $\delta_l^i$, and the MSB of $u_l^i$, $u_l^i[n-1]$ as inputs. It generates the $i_{th}$ error of layer $l-1$, $\delta_{l-1}^i$. A 3-bit TFHE-based backward *iReLU* unit is shown in Figure 2(b), where we first compute the negation of the MSB of $u_l^i$, $\overline{u_l^i[2]}$. We then compute each bit of $\delta_{l-1}^i$ by ANDing each bit of $\delta_l^i$ with $\overline{u_l^i[2]}$. If $u_l^i[2] = 0$, $\delta_{l-1}^i = \delta_l^i$; otherwise $\delta_{l-1}^i = 0$. An $n$-bit backward *iReLU* unit requires 1 TFHE NOT gate without bootstrapping and $n-1$ TFHE AND gates with bootstrapping. Our TFHE-based forward or backward *ReLU* function takes only 0.1 second, while a BGV-lookup-table-based activation consumes 307.9 seconds on our CPU baseline.

**Forward Softmax**. A *softmax* operation takes $n$ $u_l^i$s as its input and normalizes them into a probability distribution consisting of $n$ probabilities proportional to the exponentials of inputs. The *softmax* activation can be described as: $softmax(u_l^i) = d_l^i = \frac{e^{u_l^i}}{\Sigma_i e^{u_l^i}}$. We use TFHE homomorphic multiplexers to implement a 3-bit *softmax* unit shown in Figure 2(c), where we have 8 entries denoted as $S_0 \sim S_7$ for a 3-bit TFHE-lookup-table-based exponentiation unit in *softmax*; and each entry has 3-bit. The $i_{th}$ neuron $u_l^i$ is used to look up one of the eight entries, and the output is $e^{u_l^i}$, and *softmax* unit $d_l^i$ can be further obtained by BGV additions and division. There are two TFHE gates with bootstrapping on the critical path of each TFHE homomorphic multiplexer. An $n$-bit *softmax* unit requires $2^n$ TFHE gates with bootstrapping. Compared to BGV-lookup-table-based *softmax*, our TFHE-based *softmax* unit reduces the activation latency from 307.9 seconds to only 3.3 seconds.

**Backward Softmax**. To efficiently back-propagate the loss of *softmax*, we adopt the derivative of quadratic loss function described as: $isoftmax(d_l^i, t^i) = \delta_l^i = d_l^i - t^i$, where $t^i$ is the $i_{th}$ ground truth. The quadratic loss function requires only homomorphic multiplications and additions. Although it is feasible to implement the quadratic loss function by TFHE, when considering the switching overhead from BGV to TFHE, we use BGV to implement the quadratic loss function.

**Pooling**. It is faster to adopt TFHE to implement *max pooling* operations. But considering the switching overhead from BGV to TFHE, we adopt BGV to implement *average pooling* operations requiring only homomorphic additions and multiplications.

### 3.2 Switching between BGV and TFHE

BGV can efficiently process vectorized arithmetic operations, while TFHE runs logic operations faster. During private training, we plan to use BGV for convolutional, fully-connected, average pooling, and batch normalization layers, and adopt TFHE for activation operations. To use both BGV and TFHE, we propose a cryptosystem switching technique switching Glyph between BGV and TFHE cryptosystems.

Both BGV and TFHE are built on Ring-LWE [8, 6], but they cannot naïvely switch between each other. Because BGV and TFHE work on different plaintext spaces. The plaintext space of BGV is the ring $\mathcal{R}_p = \mathbb{Z}[X]/(X^N+1) \mod p^r$, where $p$ is a prime and $r$ is an integer. We denote the BGV plaintext space as $\mathbb{Z}_N[X] \mod p^r$. TFHE has three plaintext spaces [9] including *TLWE*, *TRLWE* and *TRGSW*. TLWE encodes individual continuous plaintexts over the torus $\mathbb{T} = \mathbb{R}/\mathbb{Z} \mod 1$. TRLWE encodes continuous plaintexts over $\mathbb{R}[X] \mod (X^N + 1) \mod 1$. We denote the TRLWE plaintext space as $\mathbb{T}_N[X] \mod 1$, which can be viewed as the packing of $N$ individual coefficients. TRGSW

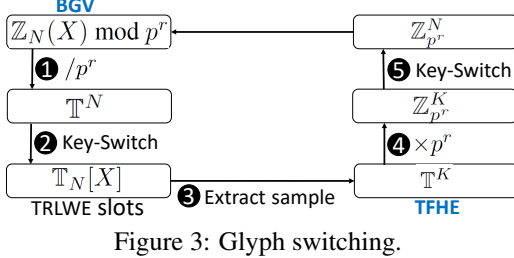

Figure 3: Glyph switching.

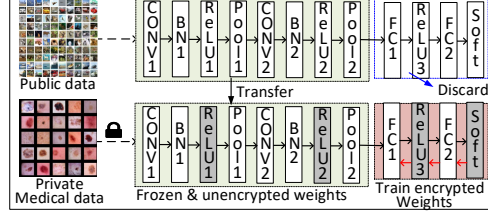

Figure 4: Transferring unencrypted features.

encodes integer polynomials in $\mathbb{Z}_N[X]$ with bounded norm. Through key-switching, TFHE can switch between these three plaintext spaces. Our cryptosystem switching scheme maps the plaintext spaces of BGV and TFHE to a common algebraic structure using natural algebraic homomorphisms. The cryptosystem switching can happen on the common algebraic structure.

Our cryptosystem can enable Glyph to use both TFHE and BGV cryptosystems by homomorphically switching between different plaintext spaces, as shown in Figure 3.

- **From BGV to TFHE**. The switch from BGV to TFHE homomorphically transforms the ciphertext of $N$ BGV slots encrypting $N$ plaintexts over $\mathbb{Z}_N[X] \mod p^r$ to $K$ TLWE-mode TFHE ciphertexts, each of which encrypts plaintexts over $\mathbb{T} = \mathbb{R}/\mathbb{Z} \mod 1$. The switch from BGV to TFHE includes three steps. ❶ Based on Lemma 1 in [14], $\mathbb{Z}_N[X] \mod p^r$ homomorphically multiplying $p^{-r}$ is a $\mathbb{Z}_N[X]$-module isomorphism from $\mathcal{R}_p = \mathbb{Z}_N[X] \mod p^r$ to the submodule of $\mathbb{T}_N[X]$ generated by $p^{-r}$. Via multiplying $p^{-r}$, we can convert integer coefficients in the plaintext space of BGV into a subset of torus $\mathbb{T}$ consisting of multiples of $p^{-r}$. In this way, we extract $N$ coefficients from the BGV plaintexts over $\mathbb{Z}_N[X] \mod p^r$ to form $\mathbb{T}^N$. ❷ Based on Theorem 2 in [14], we use the functional key-switching to homomorphically convert $\mathbb{T}^N$ into $\mathbb{T}_N[X]$, which is the plaintext space of the TRLWE-mode of TFHE. ❸ We adopt the SampleExtract function [14] of TFHE to homomorphically achieve $K$ individual TLWE ciphertexts from $\mathbb{T}_N[X]$. Given a TRLWE ciphertext $c$ of a plaintext $\mu$, *SampleExtract(c)* extracts from $c$ the TLWE sample that encrypts the $i_{th}$ coefficient $\mu_i$ with at most the same noise variance or amplitude as $c$. To further support binary gate operations, $K \times M$ functional bootstrappings [14] are required for the conversion between integer-based plaintext $\mathbb{T}^K$ and binary-based plaintext $\mathbb{T}^{K \times M}$, where $M$ is the bitwidth of a slot number in BGV operations.
- **From TFHE to BGV**. The switch from TFHE to BGV is to homomorphically transform $K$ TFHE ciphertexts in the TLWE-mode $(m_0, m_1, \ldots, m_{K-1})$ in $\mathbb{T}^K$ to a BGV $N$-slot ciphertext whose plaintexts are over $\mathbb{Z}_N[X] \mod p^r$. ❹ Based on Theorem 3 in [14], we can use the functional gate bootstrapping of TFHE to restrict the plaintext space of TFHE in the TLWE-mode to an integer domain $\mathbb{Z}_{p^r}^K$ consisting of multiples of $p^{-r}$. ❺ The plaintext space transformation from $\mathbb{Z}_{p^r}^K$ to $\mathbb{Z}_{p^r}^N$ is a $\mathbb{Z}_N[X]$-module isomorphism, so we can also use the key-switching to implement it. At last, the BGV $N$-slot ciphertext whose plaintexts are over $\mathbb{Z}_N[X] \mod p^r$ is obtained. And homomorphic additions and multiplications of BGV can be implemented by TFHE operations.

### 3.3 Transfer Learning for Private DNN Training

Although FHESGD [2] shows that it is feasible to homomorphically train a 3-layer MLP, it is still very challenging to homomorphically train a convolutional neural network (CNN), because of huge computing overhead of homomorphic convolutions. We propose to apply transfer learning to reduce computing overhead of homomorphic convolutions in private CNN training. Although several prior works [15, 16] adopt transfer learning in privacy-preserving inferences, to our best knowledge, this is the first work to use transfer learning in private training.

Transfer learning [17, 18, 19] can reuse knowledge among different datasets in the same CNN architecture, since the first several convolutional layers of a CNN extracts general features independent of datasets. Applying transfer learning in private training brings two benefits. First, transfer learning reduces the number of trainable layers, i.e., weights in convolutional layers are fixed, so that training latency can be greatly reduced. Second, we can convert computationally expensive convolutions between ciphertext and ciphertext to cheaper convolutions between ciphertext and plaintext, because the fixed weights in convolutional layers are not updated by encrypted weight gradients. Moreover, transfer learning does not hurt the security of FHE-based training, since the input, activations, losses and gradients are still encrypted.

We show an example of applying transfer learning in private CNN training in Figure 4. We reuse the first two convolutional layers trained by unencrypted CIFAR-10, and replace the last two fully-connected layers by two randomly initialized fully-connected layers, when homomorphically training of the same CNN architecture on an encrypted skin cancer dataset [20]. During private training on the skin cancer dataset, we update weights only in the last two fully-connected layers. In this way, the privacy-preserving model can reuse general features learned from public unencrypted datasets. Meanwhile, in private training, computations on the first several convolutional and batch normalization layers are computationally cheap, since their weights are fixed and unencrypted.

## 4 Experimental Methodology

**Cryptosystem Setting**. For BGV, we used the same parameter setting rule as [21], and the HElib [7] library to implement all related algorithms. We adopted the $m_{th}$ cyclotomic ring with $m = 2^{10} - 1$, corresponding to lattices of dimension $\psi(m) = 600$. This native plaintext space has 60 plaintext slots which can pack 60 input ciphertexts. The BGV setting parameters yield a security level of $> 80$ bits. Both BGV and TFHE implement bootstrapping operations and support fully homomorphic encryption. We set the parameters of TFHE to the same security level as BGV, and used the TFHE [9] library to implement all related algorithms. TFHE is a three-level scheme. For first-level TLWE, we set the minimal noise standard variation to $\underline{\alpha} = 6.10 \cdot 10^{-5}$ and the count of coefficients to $\underline{n} = 280$ to achieve the security level of $\underline{\lambda} = 80$. The second level TRLWE configures the minimal noise standard variation to $\alpha = 3.29 \cdot 10^{-10}$, the count of coefficients to $n = 800$, and the security degree to $\lambda = 128$. The third-level TRGSW sets the minimal noise standard variation to $\overline{\alpha} = 1.42 \cdot 10^{-10}$, the count of coefficients to $\overline{n} = 1024$, the security degree to $\overline{\lambda} = 156$. We adopted the same key-switching and extract-sample parameters of TFHE as [14].

**Simulation, Dataset and Network Architecture**. We evaluated all schemes on an Intel Xeon E7-8890 v4 2.2GHz CPU with 256GB DRAM. It has two sockets, each of which owns 12 cores and supports 24 threads. Our encrypted datasets include MNIST [22] and Skin-Cancer-MNIST [20]. Skin-Cancer-MNIST consists of 10015 dermatoscopic images and includes a representative collection of 7 important diagnostic categories in the realm of pigmented lesions. We grouped it into a 8K training dataset and a 2K test dataset. We also used SVHN [23] and CIFAR-10 [24] to pre-train our models which are for transfer learning on encrypted datasets. We adopted two network architectures, a 3-layer MLP [2] and a 4-layer CNN shown in Figure 4. The 3-layer MLP has a $28 \times 28$ input layer, a 128-neuron hidden layer and a 32-neuron hidden layer. The CNN includes two convolutional layers, two batch normalization layers, two pooling layers, three *ReLU* layers and two fully-connected layers. The CNN architectures are different for MNIST and Skin-Cancer-MNIST. For MNIST, the input size is $28 \times 28$. There are $6 \times 3 \times 3$ and $16 \times 3 \times 3$ weight kernels, respectively, in two convolutional layers. Two fully connected layers have 84 neurons and 10 neurons respectively. For Skin-Cancer-MNIST, the input size is $28 \times 28 \times 3$. There are $64 \times 3 \times 3 \times 3$ and $96 \times 64 \times 3 \times 3$ weight kernels in two convolutional layers, respectively. Two fully-connected layers are 128 neurons and 7 neurons, respectively. We quantized the inputs, weights and activations of two network architectures with 8-bit by the training quantization technique in SWALP [25].

## 5 Results and Analysis

### 5.1 MNIST

**FHESGD**. During a mini-batch, the 3-layer FHESGD-based MLP [2] is trained with 60 MNIST images. Each BGV lookup-table operation consumes 307.9 seconds, while a single BGV MAC operation costs only 0.012 seconds. Although activation layers of FHESGD require only a small number of BGV lookup-table operations, they consumes 98% of total training latency. The FHESGD-based MLP makes all homomorphic multiplications happen between ciphertext and ciphertext (MultCC), though homomorphic multiplications between ciphertext and plaintext (MultCP) are computationally cheaper. The total training latency of a 3-layer FHESGD-based MLP for a mini-batch is **118K** seconds, which is about 1.35 days [2].

**TFHE Activation and Cryptosystem Switching**. We replace all activations of the 3-layer FHESGD-based MLP by our TFHE-based *ReLU* and *softmax* activations, and build it as a Glyph-based MLP. We also integrate our cryptosystem switching into the Glyph-based MLP to perform homomorphic MAC operations by BGV, and conduct activations by TFHE. The mini-batch training latency breakdown of the 3-layer Glyph-based MLP on a single CPU core is shown in Table 3(a). Because of the logic-operation-friendly TFHE, the processing latency of activation layers of Glyph significantly

**(a) Glyph-based MLP.**

| Layer | BGV-TFHE (s) | BFV-TFHE (s) | HOP # | Mul-tCC # | Ad-dCC # | Act # | Sw-itch |
|---|---|---|---|---|---|---|---|
| FC1-f | 1.37K | 4.3K | 201K | 100K | 100K | 0 | B-T |
| Act1-f | 19.2 | 19.2 | 128 | 0 | 0 | 128 | T-B |
| FC2-f | 57.1 | 178.2 | 8.2K | 4.1K | 4.1K | 0 | B-T |
| Act2-f | 4.82 | 5.3 | 32 | 0 | 0 | 32 | T-B |
| FC3-f | 6.02 | 14.2 | 640 | 320 | 320 | 0 | B-T |
| Act3-f | 34.76 | 3079 | 10 | 0 | 0 | 10 | T-B |
| Act3-e | 0.1 | 0.1 | 10 | 0 | 0 | 0 | - |
| FC3-e | 4.32 | 13.79 | 640 | 320 | 320 | 0 | - |
| FC3-g | 6.02 | 15.4 | 640 | 320 | 320 | 0 | B-T |
| Act2-e | 4.82 | 4.82 | 32 | 0 | 0 | 32 | T-B |
| FC2-e | 55.4 | 176.5 | 8.2K | 4.1K | 4.1K | 0 | - |
| FC2-g | 62.1 | 183.2 | 8.2K | 4.1K | 4.1K | 0 | B-T |
| Act1-e | 19.2 | 19.2 | 128 | 0 | 0 | 128 | T-B |
| FC1-g | 1.3K | 4.3K | 201K | 100K | 100K | 0 | - |
| Total | 2.9K | 12.3K | 429K | 213K | 21K | 330 | - |

**(b) Glyph-based CNN.**

| Layer | BGV-TFHE (s) | BFV-TFHE (s) | HOP # | **Mul-tCP #** | Mul-tCC # | Ad-dCC # | Act # | Swi-tch |
|---|---|---|---|---|---|---|---|---|
| Con1-f | 69 | 226 | 73K | 37K | 0 | 37K | 0 | - |
| BN1-f | 61 | 106 | 15K | 8K | 0 | 8K | 0 | B-T |
| Act1-f | 321 | 321 | 4.1K | 0 | 0 | 0 | 4.1K | T-B |
| Pool1-f | 17 | 56 | 18K | 9.1K | 0 | 9.1K | 0 | - |
| Conv2-f | 33 | 104 | 35K | 17K | 0 | 17K | 0 | - |
| BN2-f | 27 | 43 | 7K | 3K | 0 | 3K | 0 | B-T |
| Act2-f | 151 | 151 | 1.9K | 0 | 0 | 0 | 1.9K | T-B |
| Pool2-f | 7 | 22 | 7.2K | 3.6K | 0 | 3.6K | 84 | - |
| FC1-f | 228 | 1.4K | 67K | 0 | 34K | 34K | 0 | B-T |
| Act3-f | 8.2 | 8.2 | 84 | 0 | 0 | 0 | 84 | T-B |
| FC2-f | 6.1 | 36.3 | 1.68K | 0 | 840 | 840 | 0 | B-T |
| Act4-f | 68 | 68.6 | 10 | 0 | 0 | 0 | 10 | T-B |
| Act4-e | 0.1 | 0.1 | 10 | 0 | 0 | 10 | 0 | - |
| FC2-e | 6 | 36.2 | 1.68K | 0 | 840 | 840 | 0 | - |
| FC2-g | 31 | 61.2 | 1.68K | 0 | 840 | 840 | 0 | B-T |
| Act3-e | 32 | 32 | 84 | 0 | 0 | 0 | 84 | T-B |
| FC1-g | 227 | 1.4K | 67K | 0 | 34K | 34K | 0 | - |
| Total | 1.3K | 4.2K | 1716K | 746K | 106K | 852K | 14K | - |

Table 3: The mini-batch training latency comparison. **BGV-TFHE** is the latency of Glyph implementing linear layers by BGV and non-linear layers by TFHE. **BFV-TFHE** is the latency of Chimera [14] implementing linear layers by BFV and non-linear layers by TFHE. **HOP** includes the number of homomorphic operations. **MultCC** indicates the number of multiplications between ciphertext and ciphertext. **MultCP** means the number of multiplications between ciphertext and plaintext. **AddCC** is the number of additions between ciphertext and ciphertext. **Switch** means the cryptosystem switching. **FC** is a fully-connected layer. **Act** denotes an activation layer. **BN** is a batch normalization layer. **Pool** denotes an average pooling layer. **N-f** means a N layer in forward propagation. **N-e** is the error computation of a N layer in backward propagation. **N-g** is the gradient computation of a N layer in backward propagation. **B-T** indicates BFV/BGV switches to TFHE. And **T-B** indicates TFHE switches to BFV/BGV.

decreases. The cryptosystem switching introduces only small computing overhead. For instance, compared to the counterpart in the FHESGD-based MLP, *FC1-f* increases processing latency by only 4.9%, due to cryptosystem switching overhead. Because of fast activations, compared to the FHESGD-based MLP, our Glyph-based MLP reduces mini-batch training latency by 97.4% but maintains the same test accuracy. The MLP can also be implemented by a recent cryptosystem switching technique Chimera [14], where linear layers are built upon BFV and nonlinear layers depend on TFHE. Because of faster BGV MultCCs, as Table 3(a) shows, our Glyph-based MLP (BGV-TFHE) decreases mini-batch training latency by 76.4% over Chimera (BFV-TFHE).

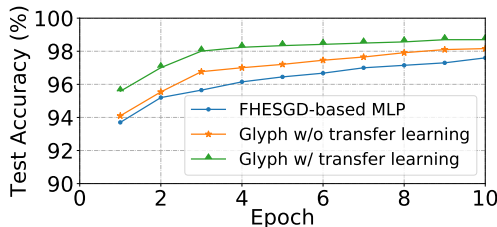

Figure 5: Accuracy comparison on MNIST.

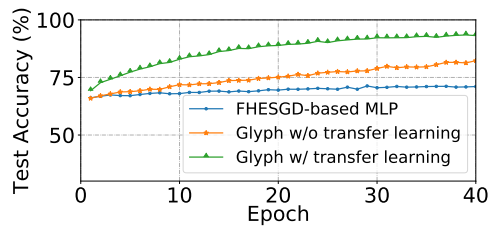

Figure 6: Accuracy comparison on Skin-Cancer.

**Transfer Learning on CNN**. We use our TFHE-based activations and cryptosystem switching to build a Glyph-based CNN, whose detailed architecture is explained in Section 4. We implement transfer learning in the Glyph-based CNN by fixing convolutional layers trained by SVHN and training only two fully-connected layers. The mini-batch training latency breakdown of the Glyph-based CNN with transfer learning on a single CPU core is shown in Table 3(b). Because the weights of convolutional layers are unencrypted and fixed, our Glyph-based CNN significantly reduces the number of MultCCs, and adds only computationally cheap MultCPs. The Glyph-based CNN decreases training latency by 56.7%, but improves test accuracy by ~2% over the Glyph-based

MLP. If we use Chimera to implement the same CNN, our Glyph-based CNN (BGV-TFHE) reduces training latency by 69% over Chimera (BFV-TFHE), due to much faster BGV MultCCs and MultCPs.

**Test Accuracy**. The test accuracy comparison of the FHESGD-based MLP and the Glyph-based CNN is shown in Figure 5, where all networks are trained in the plaintext domain. It takes 5 epochs for the FHESGD-based MLP to reach 96.4% test accuracy on MNIST. After 5 epochs, the Glyph-based CNN can achieve 97.1% test accuracy even without transfer learning. By reusing low-level features of the SVHN dataset, the Glyph-based CNN with transfer learning obtains 98.6% test accuracy. The CNN architecture and transfer learning particularly can help the FHE-based privacy-preserving DNN training to achieve higher test accuracy when we do not have long time for training.

## 5.2 Skin-Cancer-MNIST

We built a Glyph-based MLP and a Glyph-based CNN for Skin-Cancer-MNIST by our TFHE-based activations, cryptosystem switching and transfer learning. The reductions of mini-batch training latency of Skin-Cancer-MNIST are similar to those on MNIST. The test accuracy comparison of the FHESGD-based MLP and the Glyph-based CNN is shown in Figure 6. For transfer learning, we first train the Glyph-based CNN with CIFAR-10, fix its convolutional layers, and then train its fully-connected layers with Skin-Cancer-MNIST. On such a more complex dataset, compared to the FHESGD-based MLP, the Glyph-based CNN without transfer learning increases training accuracy by 2% at the $15_{th}$ epoch. The transfer learning further improves test accuracy of the Glyph-based CNN to 73.2%, i.e., a 4% test accuracy boost. TFHE-based activations, cryptosystem switching and transfer learning makes Glyph efficiently support deep CNNs.

Table 4: The comparison of overall training latency of Glyph.

| Dataset | Name | Thread # | Mini-batch | Epoch # | Time | Acc(%) |
|---|---|---|---|---|---|---|
| MNIST | FHESGD | 48 | 2.3 hours | 50 | 13.4 years | 97.8 |
| | Chimera | 48 | 0.14 hours | 5 | 28.6 days | 98.6 |
| | Glyph | 48 | 0.04 hours | 5 | **8 days** | **98.6** |
| Cancer | FHESGD | 48 | 2.4 hours | 30 | 1.1 years | 70.2 |
| | Chimera | 48 | 0.29 hours | 15 | 25.1 days | 73.2 |
| | Glyph | 48 | 0.08 hours | 15 | **7 days** | **73.2** |

## 5.3 Overall Training Latency

The overall training latency of multiple threads on our CPU baseline is shown in Table 4. We measured the mini-batch training latency by running various FHE-based training for a mini-batch. *We estimated the total training latency via the product of the mini-batch training latency and the total mini-batch number for a training.* For MNIST, training the MLP requires 50 epochs, each of which includes 1000 mini-batches (60 images), to obtain 97.8% test accuracy. Training the Glyph (BGV-TFHE)-based CNN on MNIST requires only 5 epochs to achieve 98.6% test accuracy. The overall training latency of the CNN is 8 days. Although a Chimera (BFV-TFHE)-based CNN can also achieve 98.6% test accuracy, its training requires 28.6 days, $2.6\times$ slower than Glyph. For Skin-Cancer-MNIST, it takes 30 epochs, each of which includes 134 mini-batches. Training the Chimera-based or Glyph-based CNN requires only 15 epochs to obtain 73.2% test accuracy. By 48 threads, the training of the Chimera-based CNN can be completed within 26 days. In contrast, training the Glyph-based CNN requires only 7 days.

## 6 Conclusion

In this paper, we propose, Glyph, a FHE-based privacy-preserving technique to fast and accurately train DNNs on encrypted data. Glyph performs *ReLU* and *softmax* by logic-operation-friendly TFHE, while conducts MAC operations by vectorial-arithmetic-friendly BGV. We create a cryptosystem switching technique to switch Glyph between TFHE and BGV. We further apply transfer learning on Glyph to support CNN architectures and reduce the number of homomorphic multiplications between ciphertext and ciphertext. Our experimental results show Glyph obtains state-of-the-art accuracy, and reduces training latency by $69\% \sim 99\%$ over prior FHE-based privacy-preserving techniques on encrypted datasets.

## Broader Impact

In this paper, we propose a FHE-based privacy-preserving technique to fast and accurately train DNNs on encrypted data. Average users, who have to rely on big data companies but do not trust them, can benefit from this research, since they can upload only their encrypted data to untrusted servers. No

one may be put at disadvantage from this research. If our proposed technique fails, everything will go back to the state-of-the-art, i.e., untrusted servers may leak sensitive data of average users.

## Acknowledges

The authors would like to thank the anonymous reviewers for their valuable comments and helpful suggestions. This work was partially supported by the National Science Foundation (NSF) through awards CIF21 DIBBS 1443054, CINES 1835598, CCF-1908992 and CCF-1909509.

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
