[Reviews · NeurIPS 2020]

Review 1

Summary and Contributions: The paper describes an FHE based technique for privacy-preserving training neural networks. The core contributions of this work are a new cryptographic scheme that allows for switching between the BGV and TFHE cryptosystems since the both are suited for different types of layers of a neural network. This new technique offers ~3x speed up over the state-of-art Chimera approach

Strengths: The main strength of this work is the implementation of a newly proposed scheme for switching between BGV and TFHE ciphertexts. This scheme is derived from the work on Chimera (which switches between BFV and TFHE ciphertexts). The authors perform detailed benchmarking of the runtime of linear layers in a neural network and show that the BGV is a better choice when compared with BFV. Based on this insight they implement their new scheme for two networks and show that they can reduce latency by ~69%. Finally, the authors use transfer learning to make the task of training competitive CNNs more tractable

Weaknesses: The main weakness of the work stem from its incremental nature. Very similar ideas have been demonstrated in the past (e.g. Chimera for cryptosystem switching). The notion of using transfer learning for training although novel is not a substantial leap in the context of past work like EPIC. Finally, there is the issue with the scalability of these approaches to real world problems. Training a network for MNIST/CIFAR-10 like task after using transfer learning still takes >7days. The authors should ideally compare the same with plaintext and/or SGX based approaches to give a rough sense of the computational overhead.

Correctness: The technical claims in the paper seem fair and borne out by the experimental results. The authors claim 69-99% reduction in latency in the abstract. The 69% number is explicitly reiterated in the main text however the exact conditions for the 99% claim are unclear. It would be good to highlight this condition more explicitly

Clarity: The paper is well written. It does require prior familiarity with FHE cryptosystems but that is well justified given the technical nature of the contributions. The bitwidths used to represent the various activations could be specfied a bit more clearly (e.g. Section 3 shows bitwidth 3 LUTs but Section 5 indicates 8-bit quantization)

Relation to Prior Work: The authors clearly highlight that their contribution differs from prior art in terms of the selection of BGV vs BFV for the linear layers. Similarly the use of transfer learning for training is clearly identified.

Reproducibility: Yes

Additional Feedback:


Review 2

Summary and Contributions: The paper presents a new method for training neural networks on homomorphically encrypted data. The technical contributions over prior work are (1) using a combination of BGV and TFHE schemes for delegating parts of the computation to the most appropriate schemes; (2) implementing a new TFHE-BGV scheme switching method. The paper demonstrates significant performance improvements over prior work.

Strengths: The paper is beautifully written, at least from the point of view of a reader who is familiar with homomorphic encryption. All claims are easy to follow and mostly well justified; comparisons to prior work are clear and thorough. This work is likely to be of interest to ML privacy audiences.

Weaknesses: I'm not sure how readable this is by people unfamiliar with homomorphic encryption. Unfortunately, with the low page limit, it just may be that such material is not possible to present at NeurIPS, and/or is simply outside the scope of the conference. There are a couple of issues I have with the paper: - I didn't see data sizes presented anywhere. How large is the encrypted training data, and how large are the key switching keys? - The machine used is very powerful. What was the memory use of the implementation? Is there a chance to run this on a weaker machine? If not, is it purely an implementation issue of the libraries used? - Can you comment on the machines used to evaluate the prior work, and how those machines may compare to your setup? - You mention that HEAAN supports floating point computations better. Is there a reason it was not used, instead of BGV? - Regarding "Broader impact", I would say that one disadvantage of training on encrypted data is that when data is contributed by multiple sources (through public-key encryption) any kind of model poisoning may be impossible to detect during the training phase. Have you considered such issues?

Correctness: The claims, methodology, and comparison to prior work in the paper seem both correct and meaningful.

Clarity: The paper is very clear and excellently written for homomorphic encryption experts, but I'm not sure how others will read it.

Relation to Prior Work: The paper contains a thorough comparison comparison to prior work, but I could not find anything about the machines used for evaluating the performance of said prior work.

Reproducibility: No

Additional Feedback: There is a typo "transferring learning" repeated multiple times.


Review 3

Summary and Contributions: The paper proposes a FHE-based technique, Glyph, to train DNNs fast and accurately on encrypted data. Their proposed method switches TFHE and BGV cryptosystems using logic operation-friendly TFHE. Their method achieves SOTA performance. The paper claim that their method is the first work to use transfer learning in private training.

Strengths: The paper is the first to apply transfer learning into training DNNs on encrypted data. The proposed system balances speed and accuracy. The performance is convincing and achieves state-of-the-art performance.

Weaknesses: 1. All components, BGV and TFHE, are borrowed from other papers. Similar to Chimera, the switching mechanism is not proposed by the authors. Even transfer learning has developed for a long time. The authors are suspect of just ensembling these ideas together. The reviewer doubts the novelty of the proposed method. 2. The authors did not explain why FHESGD equipped with BGV performs worse than their system equipped with TFHE-BGV because the authors claim that the BGV is better than TFHE. The reviewer expects more analysis of mechanics. Minor: Fig. 5 and Fig.6 are supposed to compare with Chimera.

Correctness: The claims and method are basically correct. However, the reviewer has some concerns: 1. Why do not use HEAAN? Please give some explanation. 2. The switching strategy is BGV firstly and TFHE latter. But why do the authors take this order?

Clarity: The paper is well written. But please pay some attention in layout. Because it is hard to read figures/tables and the corresponding text are far away.

Relation to Prior Work: Very clearly. And explain the motivation for choosing the key components well.

Reproducibility: No

Additional Feedback: Try to propose a novel mechanism for training DNNs on encrypted data.

[Author Response · NeurIPS 2020]

We thank the reviewers for their careful reading of the manuscript and their constructive suggestions.

**Reviewer-1/3, Novelty of switching & comparing against Chimera**: We compared Glyph against Chimera [14]. Chimera supports the switching between BFV and TFHE, while Glyph enables the switching between BGV and TFHE. **Chimera CANNOT support the switching between BGV and TFHE**. We selected BGV for two reasons. (1) Our baseline FHESGD [2] adopted BGV. (2) MultCPs and MultCCs of BGV are faster than those of BFV. We further demonstrated Glyph achieves faster privacy-preserving training speed than Chimera but obtains the same accuracy.

**Reviewer-1/3, Novelty of transfer learning & comparing against EPIC**: Glyph is the first work to use transfer learning to achieve fast non-interactive HE-based privacy-preserving CNN training. Based on transfer learning, EPIC replaces the last fully-connected layer of a neural network by a SVM and retrains the network by the same plaintext dataset. During an inference of EPIC, the first several layers are computed by the client, while the last layer of SVM is done by the server. EPIC depends on multi-party computation that exchanges huge amounts of data between the client and the server. Some users may not have such large network bandwidth. Moreover, EPIC CANNOT work with state-of-the-art CNNs. In contrast, Glyph first trains a CNN network model by a plaintext public dataset. And then, it homomorphically retrains the CNN model with a freshly initialized full-connected layer by an encrypted private dataset based on transfer learning. Except sending the encrypted input data, the training of Glyph does not involve the client.

**Reviewer-1/2, Scalability, computing overhead and machines**: We reported the training latency in Table 4. Compared to our baseline FHESGD [2], Glyph is more scalable, since it can support the training of deeper CNNs on larger datasets, e.g.., Skin-Cancer-MNIST. We will add the training latency on plaintext data in the next version of this manuscript. For data sizes (encrypted training data and key-switching keys), a BGV ciphertext with 60 slots is 256 KB. 60 Skin-Cancer-MNIST images cost $28 \times 28 \times 3 \times 8 \times 256KB$ = 4.6GB. The amortized size of each encrypted image is 76.6MB. One TFHE ciphertext with 1 slot is 2 KB. Each encrypted image occupies $28 \times 28 \times 3 \times 8 \times 2KB$ = 36.7 MB. Key-switching key samples have the size of 64 MB. Our baseline FHESGD uses a 2.30GHz Intel Xeon E5-2698v3 processor with two sockets and sixteen cores per socket. The machine has 250GB of main memory. And Glyph is tested on an Intel Xeon E7-8890v4 2.2GHz CPU with 256GB DRAM. The CPU also has two sockets, each of which owns 12 cores and supports 24 threads. Two machine configurations are similar. The peak main memory usage of Glyph is $\sim$150GB.

**Reviewer-1, Performance improvement 69%$\sim$99%**: Our Glyph-based CNN (BGV-TFHE) reduces the training latency by 69% over Chimera (BFV-TFHE) on the MNIST dataset. Compared to the FHESGD-based MLP, our Glyph-based MLP reduces the training latency by 97.4% on the MNIST dataset. We will update this number in our manuscript.

**Reviewer-1, Bit-width of networks** : We used 8-bit integers. We presented a 3-bit LUT in Section 3 as one example to explain the mechanism of TFHE-based activations.

**Reviewer-2/3, Why not HEAAN?**: Although HEAAN supports fixed-point numbers, we did NOT choose HEAAN for 3 reasons. (1) The training of state-of-the-art CNNs can be accurately done with only integers [R1]. (2) Our baseline FHESGD [2] uses BGV that supports only integers. (3) In order to support complex number, compared to BGV/BFV, HEAAN has only 50% batching slots, which degrades the speed of privacy-persevering training.
**[R1]** Wu, Shuang, et al. "Training and Inference with Integers in Deep Neural Networks." International Conference on Learning Representations. 2018.

**Reviewer-2, Model poisoning and boarder impact**: We used the same threat model as FHESGD [2]. In our threat model, Glyph aims to protect the privacy of clients, i.e., the input data and the output data are encrypted. We did NOT consider model poisoning in the threat model since this is a different security problem. We will consider this issue in our future work.

**Reviewer-3, Why FHESGD is worse than Glyph, and the switching strategy**: Both FHESGD and Glyph use BGV to compute linear layers. For activations, FHESGD uses BGV-based lookup tables, which is slow, as shown in Table 2. Glyph adopts TFHE to implement nonlinear activations, which is much faster, since TFHE can naturally support binary logic operations. Glyph uses BGV first, since the first layer is typically a linear layer. It switches to TFHE, since the following layer is an activation layer.

[Meta-Review · NeurIPS 2020]

Reviewers were positive about the paper overall, despite some concerns.